# Fatty Acid Profile and Dietary Value of Thigh Meat of Broiler Chickens Receiving Mineral or Organic Forms of Zn

**DOI:** 10.3390/ani14081156

**Published:** 2024-04-11

**Authors:** Anna Winiarska-Mieczan, Małgorzata Kwiecień, Cezary Purwin, Karolina Jachimowicz-Rogowska, Marta Borsuk-Stanulewicz, Paulina Pogorzelska-Przybyłek, Bożena Kiczorowska

**Affiliations:** 1Institute of Animal Nutrition and Bromatology, University of Life Sciences in Lublin, 20-950 Lublin, Poland; malgorzata.kwiecien@up.lublin.pl (M.K.); karolina.jachimowicz@up.lublin.pl (K.J.-R.); bozena.kiczorowska@up.lublin.pl (B.K.); 2Department of Animal Nutrition and Feed Science, University of Warmia and Mazury in Olsztyn, 10-719 Olsztyn, Poland; purwin@uwm.edu.pl (C.P.); marta.borsuk@uwm.edu.pl (M.B.-S.); paulina.pogorzelska@uwm.edu.pl (P.P.-P.)

**Keywords:** Zn chelate, Zn sulphate, thigh meat, broiler chicken, fatty acid, dietary value

## Abstract

**Simple Summary:**

The way poultry are fed affects the fatty acid profile of the meat. A sufficient supply of zinc (Zn) is crucial because this element is capable of modulating fatty acid metabolism and can affect the dietary quality of meat. Our study aimed to determine whether Zn glycine chelate used in the diets of Ross 308 broiler chickens can improve the dietary quality of meat due to a beneficial modification of the fatty acid profile. Using statistical methods, we confirmed the effect of the form (sulphate vs. glycine chelate) and/or quantity of Zn used (100% or 50% of the requirement) on the fatty acid profile and dietary quality of thigh meat. However, our results imply that it is sufficient to supply any form of Zn in the amount satisfying 50% of the chicken requirements for Zn to ensure satisfactory dietary quality of their meat. Studies concerning nutritional strategies for the modification of the dietary quality of meat by altering the fatty acid profile are significant to consumers, as such modification can produce good-quality meat for safe consumption without increasing the risk of diet-related civilisational diseases and can even contribute to reducing such a risk.

**Abstract:**

The present study aimed to investigate the effect of mineral (sulphate) or organic (glycine chelate) forms of Zn used in quantities covering 100% or 50% of the Zn requirement on the fatty acid profile of the thigh muscles of Ross 308 broiler chickens. We also analysed the dietary value of this meat based on its fatty acid profile. The experimental factors did not influence the content of basic chemical components and the meat’s pH. Although, in terms of statistics, the presented study supports the statement that the form (sulphate vs. glycine chelate) and/or amount of Zn used (100% or 50% of the requirement) affects the fatty acid profile and dietary value of thigh meat, the results imply that the requirement of Ross 308 broiler chickens for Zn was also covered in full when in it was used in amounts covering 50% of the requirement, irrespective of the form in which Zn was used. However, it cannot be explicitly confirmed that the form of Zn affects the analysed parameters of thigh meat.

## 1. Introduction

The nutritional value and taste of poultry meat make it an essential part of the human diet. Particularly significant are the dietary qualities of poultry meat stemming from the low-fat and -cholesterol content and a relatively high level of polyunsaturated fatty acids (PUFAs) that are essential to ensure the correct functioning of the human body and, most importantly, protection against civilisational diseases (e.g., coronary disease, heart stroke, and autoimmune diseases) and mitigating their effects [1,2]. It is known that the way poultry is fed affects the fatty acid profile of meat [3], and it is important to avoid modifications that could have an adverse effect, particularly for thigh meat, which is higher in fat than breast meat, which is considered to be low-fat [4].

Recently, researchers have noted that poultry diets contain feed additives which oversupply minerals. Therefore, firstly, it increases the animal breeding cost, and secondly, unused minerals are excreted by animals and hence contribute to environmental pollution. Our studies [5,6] showed that, to ensure adequate growth and the slaughter and physiological performance, Ross 308 broiler chickens need zinc (Zn), iron (Fe), and copper (Cu) in amounts that are half of what Aviagen [7] recommends.

Given the dietary qualities of poultry meat, it is particularly significant to provide a sufficient supply of Zn, as the symptoms of Zn and PUFA deficiency have been observed to be similar [8], suggesting a metabolic and/or molecular relationship between them. In turn, the linoleic-to-dihomo-γ-linoleic acid ratio is a potential biomarker of Zn levels in the body, as demonstrated in studies on chickens [9]. The effect of Zn on fatty acid metabolism involves (1) participation in absorbing lipids and substances soluble in lipids; (2) the effect on prostaglandin synthesis and metabolism; (3) catalysing desaturases and elongases, transforming linoleic acid (C18:2n-6) and α-linolenic acid (C18:3n-3) into their long-chain PUFA metabolites, and palmitic acid (C16:0) into nervonic acid (C24:1n-9); and (4) the presence of superoxide dismutase (SOD), which contributes to the protection of unsaturated fatty acids against oxidation in active sites [10,11,12]. Therefore, an insufficient supply of Zn to chickens can modulate fatty acid metabolism [12]. A study involving rats demonstrated that Zn deficiency had an adverse effect on the metabolic balance between PUFAn-3 and PUFAn-6 but showed no effect of Zn deficiency on saturated fatty acids (SFAs) and monounsaturated fatty acids (MUFAs) [13]. It is worth noting that the fatty acid composition in the erythrocytes of birds and mammals is similar [14].

Poultry feed is supplemented with Zn to stimulate growth [15,16]. Previous studies on poultry have revealed that organic compounds (chelates) are absorbed better than inorganic compounds used in Poland as standards (oxides and sulphates) [5,17]. Therefore, our study assumed that Zn glycine chelate used in the diets of Ross 308 broiler chickens would improve the dietary quality of meat due to a beneficial modification of the fatty acid profile. We formulated two research objectives: (1) to verify whether the form of Zn used (mineral vs. organic) affects the chemical composition and dietary qualities of thigh meat and (2) to determine whether a reduction in Zn supply to 50% of the requirement will affect the chemical composition and dietary qualities of thigh meat. The dietary qualities of meat were analysed based on parameters such as total fat and cholesterol content; and the fatty acid profile including its indicators—the Atherogenic Index (AI), Thrombogenic Index (TI), Hypocholesterolemic/Hypercholesterolemic (H/H) fatty acid ratio, and the Health-Promoting Index (HPI).

## 2. Materials and Methods

The experiment was approved by the Local Ethics Committee for Animal Testing at the University of Natural Sciences in Lublin, Poland (Resolution No. 37/2011, on 17 May 2011). All experimental procedures complied with European Council Directive 2007/43/EC.

### 2.1. Experimental Factor

The results of our previous studies imply that feeding Ross 308 broiler chickens rations containing Zn (sulphate or chelate) covering 50% of the requirement recommended by Aviagen [7] had no adverse effect on body weight, chemical composition of liver, antioxidant status of liver and blood, biochemical and haematological parameters of blood, bone mechanical and geometric properties, bone histomorphometry, and bone turnover markers [5,6]. The reduced supply of Zn in feed (50% of the requirement) did not affect the quality and chemical composition of breast meat [11].

### 2.2. Description of the Experiment

The experiment lasted for 42 days. On the first day, two hundred one-day-old Ross 308 cocks were assigned to four experimental groups (Figure 1):−Group 1: 100-Zn-sulphate, where chickens received Zn sulphate with feed covering 100% of the requirement.−Group 2: 50-Zn-sulphate, where chickens received Zn sulphate with feed covering 50% of the requirement.−Group 3: 100-Zn-Gly, where chickens received Zn glycine chelate with feed covering 100% of the requirement.−Group 4: 50-Zn-Gly, where chickens received Zn glycine chelate with feed covering 50% of the requirement.

Each group consisted of 50 male chickens placed in 5 cages, with 10 chicks in each. The requirement for Zn was determined according to the recommendations of Ross 308 producers [7]. Feed rations (starter, 1–21 days of life; grower, 22–35 days of life; finisher, 36–42 days of life) were optimised according to the norms proposed by the National Research Council [18]. The composition and nutritional value of the experimental mixtures (Agropol, Motycz, Poland) are shown in Table 1. Chickens were placed in cages (L × W × H) containing ten birds each. The broiler chickens were raised in 1 m^2^ cages until day 42 with unlimited access to feed and water. The cages were 100 cm wide, 100 cm deep, and 75 cm high. They had replaceable grates adapted to the age of the birds and were equipped with 2 nipple drinkers and a feeder (70 cm long, 8.5 cm wide, and 9 cm deep). The cages were placed in a room at an initial temperature of 32 °C. During the experiment, the temperature in the room was reduced by 2 °C every week to finally reach 24 °C [5,6]. The lighting, humidity, and temperature were consistent with the requirements established for Ross 308 chicken rearing. The chickens were kept at a 23/1 light/dark photoperiod from day 0 to day 7; from day 8 onwards, the darkness period was on average approximately 5 h. The light intensity was 35 lux until rearing day 7 and was reduced to 5–10 lux on day 8. During the dark period, the light intensity was below 0.4 lux. The chickens were provided feed (in pellet form) and water ad libitum.

At the end of the experiment, before slaughter, all the chickens were individually weighed after 10 h of fasting, and 10 representative chickens with a body weight close to the average value in the group were decapitated after electrical stunning (150 mA, frequency of 200 Hz for 4 s). The chickens were plucked, their viscera were removed, the carcasses were stored overnight at 4 °C, and thigh muscles were collected. After removal of the skin, dissected thigh muscles were placed individually in plastic bags and frozen at −20 °C until chemical analyses were carried out (max 4 weeks).

### 2.3. Muscle Samples

On day 42 of the experiment, the chickens were slaughtered. Ten chicks from each group were randomly selected for dissection. The carcasses were cooled for 24 h at 4 °C. Subsequently, both thigh muscles were sampled from the carcasses. Deskinned, they were placed separately in plastic bags and frozen at −20 °C until chemical analyses (max 4 weeks).

### 2.4. Chemical Analyses

Before the chemical assays, the thigh meat samples were thawed at room temperature. We determined the following in the meat and feed samples.

The content of essential chemical components was determined according to AOAC [19]: crude protein (Kjeldahl method), crude fat (Soxchlet method in a Velp SER 148 unit; Velp, Usmate, Italy), and crude ash (samples ashed in a muffle furnace at 550 °C, hydrogen peroxide as oxidant).

The content of Zn in ashed samples was determined by FAAS in a Unicam 939 unit (AA Spectrometer Unicam, Shimadzu Corp., Tokyo, Japan), using Merck’s standards (Darmstadt, Germany). The content of fatty acids was determined by gas chromatography in a Varian CP-3800 GC-FID unit (Varian, Harfsen, The Netherlands). Capillary column characteristics: type of CP, WAX 52CB; DF, 0.25 mm × 60 m; helium carrier flow rate, 1.4 mL/min; column temperature, 120 °C, gradually increasing by 2 °C/min up to 210 °C; determination time, 120 min; detector FID temperature, 260 °C; other gases, hydrogen and oxygen, using Supelco 37 FAME Mix 47885-U standards (Sigma, Poznań, Poland).

Total cholesterol content was determined by colourimetry using an EPOLL 20 colourimeter and C3045 standard (Sigma, Bellefonte, PA, USA).

Muscles’ pH after 15 and 45 min from slaughter was determined in a Testo 205 unit (Testo AG, Lenzkirch, Germany), using certified buffer solutions with a pH of 4.01 and 7.0.

Accurate assay methods were described by Winiarska-Mieczan and Kwiecień [20].

All chemical analyses were performed in three replications.

### 2.5. Calculations

#### 2.5.1. Total Fatty Acid Content

Based on the content of individual fatty acids, we calculated the content of saturated fatty acids (SFAs), monounsaturated fatty acids (MUFAs), polyunsaturated fatty acids (PUFAs), unsaturated fatty acids (UFAs), PUFAn-3, and PUFAn-6. Additionally, we calculated the ratios of total PUFA to SFA, SFA to UFA, and PUFAn-6 to PUFAn-3.

#### 2.5.2. Dietary Value of Meat

Based on the fatty acid content, the following parameters were calculated: Atherogenic Index (AI), Thrombogenic Index (TI), Hypocholesterolemic-to-Hypercholesterolemic fatty acid ratio (H/H), and Health-Promoting Index (HPI) using mathematical formulas [21,22,23]:AI = (C12:0 + 4 × C14:0 + C16:0)/(∑MUFA + ∑PUFAn-6 + ∑PUFAn-3)
TI = (C14:0 + C16:0 + C18:0)/[(0.5 × ∑MUFA + 0.5 × ∑PUFAn-6 + 3 × ∑PUFAn-3 + ∑PUFAn-3)/∑PUFAn-6]
H/H = (C18:1n-9 + C18:2n-6 + C20:4n-6 + C18:3n-3 + C20:5n-3 + C22:5n-3 + C22:6n-3)/(C14:0 + C16:0)
HPI = ΣUFA/[C12:0 + (4 × C14:0) + C16:0]

Meats with high dietary value are characterised by low AI (below 1.0) and TI (below 0.5) [24,25].

#### 2.5.3. Statistical Analysis

We conducted a statistical analysis of the results using Statistica 13.1 software for Windows (TIBCO Software Inc., Palo Alto, CA, USA, https://www.statsoft.pl/statistica_13/ accessed on 2 April 2024). We calculated the arithmetic mean and standard deviation. The normal distribution of variables was assessed using the Shapiro–Wilk test; for data with a normal distribution, one-way analysis of variance (ANOVA) with planned contrasts (contrast analysis) was applied. The groups were compared based on the form (mineral vs. organic) and content of Zn (100% or 50% of the requirement). Therefore, we performed comparisons between the 100-Zn-sulphate and 100-Zn-Gly groups and between the 50-Zn-sulphate and 50-Zn-Gly groups (estimated effect of the form of Zn), as well as between the 100-Zn-sulphate and 50-Zn-sulphate groups, and between the 100-Zn-Gly and 50-Zn-Gly groups (estimating the effect of the amount of Zn used). This statistical approach allows for a reasonably reliable estimation of the significance of differences between selected groups while minimizing the increase in the α level. If the data did not follow a normal distribution, Kruskal–Wallis tests were conducted. Pearson’s correlation coefficient was also calculated between the level of Zn and the content of fatty acids and cholesterol in the thigh meat.

## 3. Results

### 3.1. Chemical Composition and pH Value of Thigh Meat

The experimental factors did not show any significant effect (*p* < 0.05) on the content of water, crude protein, crude fat, and total cholesterol in thigh meat (Table 2 and Table 3). The level and form of Zn did not modify the pH of the meat 15 and 45 min after slaughter. We found that the form of Zn affected the total content of ash (*p* = 0.023) and Zn (*p* = 0.022) in the thigh meat. The content of crude ash in meat can be presented as follows: 100-Zn-Gly > 50-Zn-Gly > 100-Zn-sulphate > 50-Zn-sulphate. The meat of chickens from the 100-Zn-Gly group contained more Zn (*p* = 0.025) than that of chickens from the 100-Zn-sulphate group. In addition, the meat of chickens in the 50-Zn-Gly group contained more Zn (*p* = 0.012) than the 50-Zn-sulphate group (Table 3).

### 3.2. Fatty Acids Profile

The experimental factors did affect (*p* < 0.05) the fatty acid profile of thigh meat, but the changes were not directional (Table 4 and Table 5). No statistically significant effect of the form and amount of Zn on total SFA was recorded (Table 5); however, such changes occurred for individual fatty acids. The highest content (*p* = 0.023) of C6:0 was recorded in the meat of chickens from the 50-Gly-Zn group (confirmed effect of the form of Zn (chelate > sulphate, *p* = 0.041) and the amount of Zn (50-Zn > 100-Zn, *p* = 0.018)). The highest level of C8:0 (*p* = 0.035), C14:0 (*p* = 0.008), and C15:0 (*p* = 0.029) fatty acids and the lowest content of C17:0 (*p* = 0.036) were found in the meat of chickens from both groups receiving chelate (statistically confirmed effect of the form of Zn). The highest (*p* < 0.05) content of C10:0 was noted in the thigh meat of chickens from the 50-Zn-sulphate group and the lowest in the 100-Zn-sulphate and 50-Zn-Gly groups. In this case, we confirmed the effect of the amount of Zn in the groups receiving sulphate (50-Zn > 100-Zn, *p* = 0.036) and that of Zn from the 50-Zn-sulphate and 50-Zn-Gly groups (sulphate > chelate, *p* = 0.037). The highest C12:0 content (*p* < 0.05) was found in the meat of chickens from groups receiving sulphate; the meat of chickens from the 50-Zn-sulphate, 100-Zn-Gly, and 50-Zn-Gly groups showed no differences (*p* < 0.05) in the content of C12:0. In this case, the effect of the Zn form was statistically confirmed only between the 100-Zn-sulphate and 100-Zn-Gly groups (sulphate > chelate, *p* = 0.017). The content of C18:0 can be represented as 50-Zn-sulphate > 100-Zn-sulphate > 100-Zn-Gly = 50-Zn-Gly (statistically confirmed effect of the form of Zn (sulphate > chelate) and the amount of Zn (50-Zn > 100-Zn in the case of sulphate).

As regards the total content of MUFA, the effect of the form of Zn was statistically confirmed, but only between the 100-Zn-sulphate and 100-Zn-Gly groups (sulphate > chelate, *p* = 0.047). There were statistically significant differences in the content of certain MUFAs: C16:1n-7, C20:1n-7, and C20:1n-9 (Table 4 and Table 5). The C16:1 n-7 content can be represented as 100-Zn-sulphate > 50-Zn-sulphate > 100-Zn-Gly = 50-Zn-Gly (statistically confirmed effect of the form of Zn (sulphate > chelate, *p* < 0.05)). The highest content of C20:1n-7 (*p* = 0.045) and C20:1n-9 (*p* = 0.038) was observed in the meat of chickens in the 50-Zn-Gly group. In the 100-Zn-sulphate and 50-Zn-sulphate groups, this level was significantly lower, whereas in the 100-Zn-Gly group, it was not significantly different from the other groups. In this case, the effect of the Zn form was statistically confirmed (chelate > sulphate), but only between the 50-Zn-sulphate and 50-Zn-Gly groups (*p* < 0.05).

No statistically significant effect of the experimental factors on the total PUFA content in chicken thigh meat was determined; however, the effect of experimental factors on the content of certain PUFA was statistically confirmed, that is, on C18:2n-6 (*p* = 0.033), C18:3n-6 (*p* = 0.020), and C20:3n-3 (*p* = 0.041) (Table 4 and Table 5). The C18:2n-6 content in meat can be represented as follows: 100-Zn-sulphate > 50-Zn-sulphate > 100-Zn-Gly = 50-Zn-Gly. The effect of the form of Zn (*p* > 0.05) and the amount of Zn were statistically confirmed, but only between the 100-Zn-sulphate and 50-Zn-sulphate groups (100-Zn > 50-Zn, *p* = 0.039). The C18:3n-6 content can be represented as follows: 50-Zn-Gly > 100-Zn-Gly > 100-Zn-sulphate = 50-Zn-sulphate. The effect of the form of Zn (chelate > sulphate, *p* > 0.05) and the amount of Zn were statistically confirmed, but only between the 100-Zn-Gly and 50-Zn-Gly groups (50-Gly > 100-Gly, *p* = 0.009). The highest (*p* < 0.05) content of C20:3n-3 was observed in the meat of chickens from the 50-Zn-Gly group, and in the 100-Zn-sulphate group, it was significantly lower. For C20:3n-3, statistically significant differences were found between the 100-Zn-sulphate and 50-Zn-Gly groups (*p* = 0.041).

The total content of UFA can be represented as 100-Zn-sulphate > 50-Zn-sulphate = 100-Zn-Gly = 50-Zn-Gly, and that of PUFAn-3 as 50-Zn-Gly > 100-Zn-Gly > 100-Zn-sulphate = 50-Zn-sulphate. We did not observe any statistically significant effects of the experimental factors (form and amount of Zn) on the content of PUFAn-6 and SFA/UFA. The total PUFA/SFA can be represented as 100-Zn-sulphate > 50-Zn-sulphate = 100-Zn-Gly = 50-Zn-Gly, whereas PUFA n-6/PUFA n-3 can be represented as 100-Zn-sulphate = 50-Zn-sulphate > 100-Zn-Gly > 50-Zn-Gly.

We observed a statistically significant correlation between the Zn content in thigh meat and the content of PUFAn-3 in meat (r = −0.959; *p* = 0.041). In contrast, such a correlation was not confirmed between the Zn content in meat and the total content of SFA, MUFA, PUFA, UFA, PUFAn-6, and total cholesterol (Table 6).

### 3.3. Dietary Value of Thigh Meat

The experimental factors had a statistically significant effect on the AI, TI, H/H, and HPI values (Table 7 and Table 8). The AI value ranged from 0.353 ± 0.04 to 0.373 ± 0.04. AI can be represented as 100-Zn-Gly > 50-Zn-Gly = 50-Zn-sulphate > 100-Zn-sulphate, and TI as 100-Zn-Gly > 50-Zn-Gly > 50-Zn-sulphate = 100-Zn-Gly. The TI ranged from 0.72 to ca. 0.74. The effects of the amount and form of Zn on TI and AI were statistically confirmed (*p* < 0.05), except for the 50-sulphate-Zn and 50-Gly-Zn group. The H/H ranged from 2.652 ± 0.30 to 2.793 ± 0.35 and can be represented as 100-Zn-sulphate > 50-Zn-Gly > 50-Zn-sulphate > 100-Zn-Gly. We obtained statistical confirmation of the effect of the amount of Zn in the feed ration (*p* < 0.05), as well as of the form in which Zn was used (*p* < 0.05). The HPI ranged from 2.71 ± 0.29 to 2.87 ± 0.34. HPI can be represented as follows: 100-Zn-sulphate > 50-Zn-sulphate > 50-Zn-Gly > 100-Zn-Gly. The HPI value was significantly dependent on the amount and form of Zn.

## 4. Discussion

The level of Zn in the body can regulate fatty acid metabolism because Zn is a cofactor of enzymes, such as desaturase and elongase, which can influence the activity of these enzymes [12,26]. Desaturation and elongation are critical processes in endogenous metabolic fatty acid pathways [9].

Studies on broiler chickens supplemented with Zn have revealed a significant effect of Zn on the transformation of endogenous fatty acids, in which a higher activity of Δ-6-desaturase was discovered [26]. Some researchers also found that Zn deficiency increased the level of PUFAn-3 at the expense of PUFAn-6 in tissue phospholipids [13]. This is associated with the involvement of Δ-6-desaturase in the transformation of PUFAn-3 fatty acids [27]. In this study, we noted a statistically significant effect of the level of Zn in meat on the content of PUFAn-3, with no significant effect on PUFAn-6, which is one of the factors suggesting that the level of Zn supplied to chickens with feed was not insufficient even when Zn sulphate covered 50% of the standard requirement. It is worth noting that the highest level of PUFAn-3 was detected in the meat of chickens receiving Zn chelate, covering 50% of the requirement, which confirms the observation that the supply of Zn to chickens was sufficient in all groups. In the case of PUFAn-6, the meat of chickens receiving chelate contained more linolenic acid and less linoleic acid than those receiving sulphate, although no statistically significant correlation was observed between the Zn and PUFAn-6 contents of meat. Given that Zn deficiency induces a decrease in the level of PUFAn-6 [28], it should be noted that the level of Zn supplied to broiler chickens was not too low.

Zn deficiency impairs long-chain PUFAs’ synthesis in the body [29]. It also increases the β-oxidation of linoleic acid, which reduces the available amount of linoleic acid required for its conversion into arachidonic acid [12,30]. We did not find a statistically significant reduction in the level of arachidonic acid in thigh meat, which additionally testifies to the sufficient supply of Zn to chickens in all the experimental groups. Furthermore, we did not observe an effect of the experimental factors on the total cholesterol content in thigh meat, and the Zn content was directly correlated with the cholesterol level [31].

The intestinal environment is crucial for Zn homeostasis in the body. Reed et al. [32] discovered that as little as four weeks of Zn deficiency had an effect on the microbiota of the caecum in broiler chickens fed Zn-deficient diets. Changes in the structure of the intestinal microbiota lead to the reduced production of short-chain fatty acids (SCFAs), which can affect Zn absorption and availability from the diet, leading to further deficiency of this mineral.

The ratio of ΣPUFA to ΣSFA in meat should exceed 0.45, which is the prerequisite of its healthiness, and values below 0.45 have a hypercholesterolemic effect on humans [11]. In the present study, the ΣPUFA/ΣSFA ratio was above 0.9 in all groups, meaning that the analysed meat had a high anticholesterolemic value. The content and mutual relationship of PUFAn-6 and PUFAn-3 determine the dietary quality of meat: PUFAn-3 determines the TI, and PUFAn-6, to the greatest extent, determines the AI [25]. Meats with high dietary value should have a low AI (below 1.0) and TI (below 0.5) and a high H/H index [25]. The AI and TI values are important because they imply the potential to stimulate blood platelet aggregation. Therefore, decreased AI and TI suggest a protective effect on coronary arteries [22]. In the presented study, in none of the groups did the AI exceed 0.4. An increased TI (>0.7) was observed, which exceeded the recommended level by approximately 40%. However, this value should not have a negative effect on the overall evaluation of the dietary value of thigh meat, as all the remaining parameters (AI, H/H, total cholesterol, and total fat) give no rise to doubt. In the 100-sulphate-Zn group, chickens were fed standard feed for broiler chickens containing Zn in the amount covering 100% of this mineral, and in that group, the TI was 0.72. In contrast, other authors have reported even higher levels of TI > 0.8, in some cases reaching ca. 1.3 [22,33]. However, one of the critical points for the TI is that PUFAn-6 should be regarded as an anti-coagulant [22]. It should be highlighted that, in the present study, the total PUFAn-6 content was not dependent on experimental factors. The H/H ratio reflects the effect of fatty acids on cholesterol metabolism; therefore, the resulting values should be as high as possible [25]. In our study, the H/H ratio of thigh meat was approximately 2.7 and 2.8. The HPI is used to evaluate the nutritional value of fat and, more accurately, fatty acids in cardiovascular diseases [22]. The HPI is the opposite of IA. In our study, the HPI in all groups was above 2.7, implying a very high degree of protection against atherogenic diseases. Contrary to AI and TI, the available literature does not provide reference values for HPI in broiler meat; this index is most often used for milk, fish, and plant products [34]. The HPI values for broiler-chicken thigh meat measured by other authors typically range from ca. 1.5 to more than 3 [22,34,35,36].

In addition, no significant effect of the experimental factors was found on the total cholesterol and total fat content of the thigh meat. Moreover, Zn chelates did not affect the crude protein content of the meat. This means that, despite its high significance, even a highly assimilable form of Zn does not considerably regulate the synthesis of proteins and fat in the body. Our previous studies did not demonstrate a significant effect of Zn glycine chelate on fat and protein contents in the meat of Ross 308 broiler chickens [11]. Nevertheless, Zn can play a role in protein synthesis at the translational level and in creating and attenuating certain peptides at the transcriptional level [37]. If the requirement for Zn is covered, its adverse effects on protein- and fat-level increases are eliminated.

## 5. Conclusions

We did not find any effect of experimental factors on the content of essential chemical components or meat pH. Although, in terms of statistics, the presented study supports the statement that the form (sulphate vs. glycine chelate) and/or amount of Zn used (100% or 50% of the requirement) affects the fatty acid profile and dietary value of thigh meat, the results imply that the requirement of Ross 308 broiler chickens for Zn was also covered in full when in it was used in amounts covering 50% of the requirement, irrespective of the form in which Zn was used. This means that, to ensure a beneficial fatty acid profile and indicators of dietary value (AI, TI, H/H, HPI, total fat, and cholesterol content) of thigh meat, the supply of Zn should cover 50% of the requirements recommended by the producers of Ross 308 chickens [7]. However, it cannot be explicitly confirmed that the form of Zn affected the analysed parameters of thigh meat. Nevertheless, the possibility of modifying and updating Zn requirements in broiler chickens should be considered.

## Figures and Tables

**Figure 1 animals-14-01156-f001:**
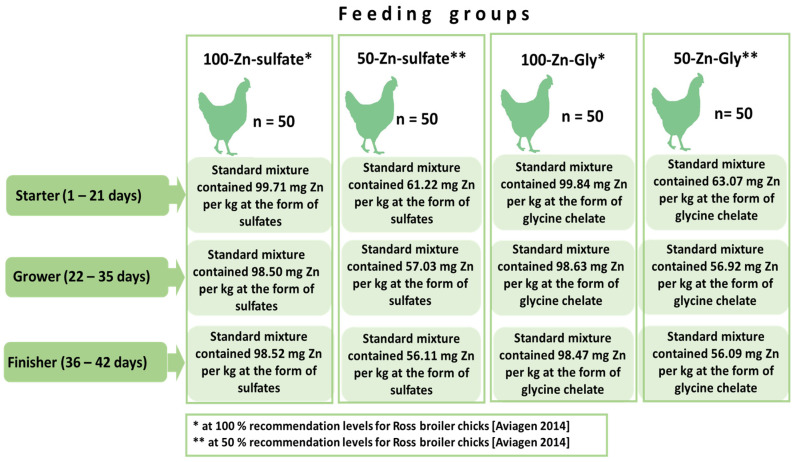
Experimental design [7].

**Table 1 animals-14-01156-t001:** Composition and nutritional value of experimental mixtures.

	Starter, 1–21 Days	Grower, 22–35 Days	Finisher, 36–42 Days
**Ingredients (g·kg^−1^)**			
Maize	244.4	400.0	400.0
Wheat	429.9	278.4	288.4
Soybean meal (46% crude protein)	250.0	249.7	228.7
Soybean oil	25.0	36.90	39.8
Monocalcium phosphate	9.00	9.00	8.10
Limestone	14.0	11.3	10.9
Sodium bicarbonate	0.80	0.80	0.80
NaCl	2.90	2.50	2.60
Vitamin–mineral premix (without Zn)	5.00	5.00	5.00
Fat–protein concentrate	10.0	-	10.0
DL-methionine 99%	3.00	2.30	2.30
L-lysine HCl	4.20	2.80	2.70
L-threonine 99%	1.80	1.30	0.70
**Nutrient value**			
*Values calculated:*			
Metabolizable energy, MJ·kg^−1^	12.7	13.1	13.2
Lysine, g·kg^−1^	12.9	11.3	10.9
Methionine + cysteine, g·kg^−1^	9.30	8.30	8.1
*Values determined:*			
Crude protein, g·kg^−1^	202	182	181
Crude fibre, g·kg^−1^	30.6	29.9	29.9
Crude fat, g·kg^−1^	46.6	60.8	64.3
Fatty acids profile, %			
Myristic (14:0)	0.03	0.07	0.07
Palmitic (16:0)	1.42	1.17	1.15
Stearic (18:0)	0.29	0.32	0.33
Oleic (18:1n-9)	2.25	2.24	2.19
Linoleic (18:2n-6)	4.72	4.95	4.96
Linolenic (18:3n-3)	1.18	0.86	0.89

**Table 2 animals-14-01156-t002:** The meat pH and chemical composition of fresh thigh meat samples.

	100-Zn-Sulphate *n* = 10	50-Zn-Sulphate *n* = 10	100-Zn-Gly *n* = 10	50-Zn-Gly *n* = 10	SEM	*p*-Value
pH_15_	6.220 ± 0.32	6.200 ± 0.44	6.221 ± 0.28	6.206 ± 0.51	1.06	0.081
pH_45_	5.431 ± 0.12	5.429 ± 0.32	5.433 ± 0.51	5.434 ± 0.39	0.52	0.113
Moisture, %	72.91 ± 3.03	72.96 ± 6.08	72.81 ± 3.76	72.98 ± 5.51	6.97	0.108
Crude ash, %	1.767 ^b^ ± 0.11	1.702 ^a^ ± 0.06	1.963 ^d^ ± 0.09	1.927 ^c^ ± 0.12	0.09	0.023
Crude protein, %	19.06 ± 1.23	19.08 ± 1.54	18.97 ± 1.22	18.84 ± 1.49	2.05	0.089
Crude fat, %	6.263 ± 2.04	6.258 ± 0.55	6.257 ± 0.49	6.253 ± 0.41	0.47	0.102
Cholesterol, mg	89.55 ± 7.56	89.48 ± 6.81	89.47 ± 6.55	89.53 ± 3.98	7.71	0.125
Zn, mg	13.67 ^a^ ± 0.94	13.70 ^a^ ± 1.01	15.87 ^b^ ± 1.34	15.79 ^b^ ± 1.44	0.12	0.022

Results are presented as the average of three repetitions; SEM—standard error of mean; pH—potential of hydrogen. ^a–d^
*p* < 0.05.

**Table 3 animals-14-01156-t003:** Effect of experimental factor on pH, and chemical composition of thigh meat samples.

	100-Zn-Sulphate vs. 50-Zn-Sulphate	100-Zn-Sulphate vs. 100-Zn-Gly	100-Zn-Gly vs. 50-Zn-Gly	50-Zn-Sulphate vs. 50-Zn-Gly
pH_15_	NS	NS	NS	NS
pH_45_	NS	NS	NS	NS
Moisture, %	NS	NS	NS	NS
Crude ash, %	0.031	0.048	0.038	0.028
Crude protein, %	NS	NS	NS	NS
Crude fat, %	NS	NS	NS	NS
Cholesterol, mg	NS	NS	NS	NS
Zn, mg	NS	0.025	NS	0.012

NS—*p* > 0.05.

**Table 4 animals-14-01156-t004:** Fatty acid profile (g/100g of total fatty acids) of thigh meat samples.

Fatty Acids		100-Zn-Sulphate *n* = 10	50-Zn-Sulphate *n* = 10	100-Zn-Gly *n* = 10	50-Zn-Gly *n* = 10	SEM	*p*-Value
6:0	Caproic	0.012 ^a^ ± 0.004	0.012 ^a^ ± 0.001	0.011 ^a^ ± 0.002	0.017 ^b^ ± 0.004	0.03	0.023
8:0	Caprylic	0.013 ^a^ ± 0.004	0.013 ^a^ ± 0.002	0.016 ^b^ ± 0.005	0.017 ^b^ ± 0.004	0.05	0.035
10:0	Capric	0.013 ^a^ ± 0.002	0.015 ^b^ ± 0.03	0.014 ^ab^ ± 0.002	0.013 ^a^ ± 0.003	0.05	0.041
12:0	Lauric	0.236 ^b^ ± 0.03	0.230 ^ab^ ± 0.01	0.224 ^a^ ± 0.02	0.222 ^a^ ± 0.03	0.09	0.047
14:0	Myristic	0.418 ^a^ ± 0.11	0.420 ^a^ ± 0.06	0.492 ^b^ ± 0.07	0.499 ^b^ ± 0.06	0.03	0.008
15:0	Pentadecanoic	0.114 ^a^ ± 0.02	0.111 ^a^ ± 0.01	0.123 ^b^ ± 0.02	0.124 ^b^ ± 0.02	0.03	0.029
16:0	Palmitic	22.26 ± 2.13	22.29 ± 1.34	22.27 ± 1.70	22.26 ± 2.27	1.67	0.077
17:0	Margaric	0.152 ^b^ ± 0.02	0.155 ^b^ ± 0.03	0.136 ^a^ ± 0.02	0.133 ^a^ ± 0.02	0.02	0.036
18:0	Stearic	6.544 ^b^ ± 0.68	6.639 ^c^ ± 0.99	6.392 ^a^ ± 0.53	6.410 ^a^ ± 0.59	0.18	0.006
20:0	Arachidic	0.144 ^b^ ± 0.05	0.138 ^ab^ ± 0.02	0.134 ^b^ ± 0.03	0.130 ^a^ ± 0.03	0.03	0.042
16:1n-7	Palmitoleic	2.681 ^b^ ± 0.35	2.682 ^b^ ± 0.43	2.563 ^a^ ± 0.67	2.567 ^a^ ± 0.65	0.45	0.039
17:1	Heptadecenoic	0.046 ± 0.02	0.044 ± 0.01	0.048 ± 0.02	0.050 ± 0.02	0.02	0.066
18:1n-9	Oleic	34.72 ± 1.34	32.87 ± 1.67	32.63 ± 2.12	33.54 ± 2.24	4.37	0.054
18:1n-11	Vaccenic	2.469 ± 1.17	2.475 ± 0.45	2.480 ± 0.52	2.264 ± 0.42	1.11	0.059
20:1n-7	Paullinic	0.073 ^a^ ± 0.02	0.073 ^a^ ± 0.01	0.087 ^ab^ ± 0.02	0.095 ^b^ ± 0.01	0.03	0.045
20:1n-9	Gondoic	0.022 ^a^ ± 0.02	0.028 ^a^ ± 0.01	0.030 ^ab^ ± 0.01	0.036 ^b^ ± 0.01	0.04	0.038
20:1n-11	Cetoleic	0.295 ± 0.05	0.305 ± 0.06	0.294 ± 0.08	0.298 ± 0.07	0.01	0.066
18:2n-6	Linoleic	25.53 ^c^ ± 1.47	25.16 ^b^ ± 1.12	24.70 ^a^ ± 1.68	24.62 ^a^ ± 1.62	2.27	0.033
20:2n-6	Eicosadienoic	0.346 ± 0.07	0.349 ± 0.07	0.345 ± 0.11	0.344 ± 0.01	0.06	0.082
18:3n-6	Linolenic	2.322 ^a^ ± 0.16	2.324 ^a^ ± 0.16	2.468 ^b^ ± 0.55	2.525 ^c^ ± 0.36	0.07	0.020
20:3n-3	Eicosatrienoic	0.164 ^a^ ± 0.04	0.173 ^ab^ ± 0.02	0.172 ^ab^ ± 0.02	0.184 ^b^ ± 0.01	0.01	0.041
20:4n-6	Arachidonic	0.118 ± 0.04	0.117 ± 0.02	0.116 ± 0.03	0.119 ± 0.03	0.02	0.076
ΣSFA		29.85 ± 2.07	30.02 ± 2.08	29.81 ± 1.98	29.82 ± 2.48	1.77	0.089
ΣMUFA		40.26 ^b^ ± 1.41	38.43 ^ab^ ± 1.48	38.09 ^a^ ± 1.88	38.80 ^ab^ ± 1.95	1.04	0.043
ΣPUFA		28.48 ± 1.55	28.12 ± 1.12	27.80 ± 1.68	27.79 ± 1.82	0.55	0.062
ΣUFA		68.74 ^b^ ± 2.22	66.55 ^a^ ± 1.78	65.89 ^a^ ± 3.01	66.58 ^a^ ± 2.83	6.04	0.035
ΣPUFAn-3		2.486 ^a^ ± 0.18	2.497 ^a^ ± 0.17	2.640 ^b^ ± 0.53	2.709 ^c^ ± 0.35	0.03	0.020
ΣPUFAn-6		25.99 ± 1.47	25.63 ± 1.10	25.16 ± 1.67	25.08 ± 1.62	0.11	0.059
ΣPUFA/SFA		0.960 ^b^ ± 0.11	0.941 ^a^ ± 0.08	0.938 ^a^ ± 0.11	0.941 ^a^ ± 0.13	0.03	0.009
ΣSFA/UFA		0.426 ± 0.04	0.437 ± 0.04	0.444 ± 0.05	0.440 ± 0.05	0.03	0.067
n-6/n-3		10.49 ^c^ ± 0.77	10.30 ^c^ ± 0.85	10.04 ^b^ ± 3.04	9.360 ^a^ ± 1.04	0.10	0.033

Results are presented as the average of three repetitions; ^a–c^ means with different superscripts in lines differ at *p* < 0.05; SEM—standard error of mean; SFA—saturated fatty acid; MUFA—monounsaturated fatty acid; PUFA—polyunsaturated fatty acid; UFA—unsaturated fatty acid.

**Table 5 animals-14-01156-t005:** Effect of experimental factors on fatty acid profile of thigh meat.

		100-Zn-Sulphate vs. 50-Zn-Sulphate	100-Zn-Sulphate vs. 100-Zn-Gly	100-Zn-Gly vs. 50-Zn-Gly	50-Zn-Sulphate vs. 50-Zn-Gly
6:0	Caproic	NS	NS	0.018	0.041
8:0	Caprylic	NS	NS	NS	0.027
10:0	Capric	0.036	NS	NS	0.037
12:0	Lauric	NS	0.017	NS	NS
14:0	Myristic	NS	0.009	NS	0.040
15:0	Pentadecanoic	NS	0.029	NS	0.044
16:0	Palmitic	NS	NS	NS	NS
17:0	Margaric	NS	0.033	NS	0.024
18:0	Stearic	0.029	0.027	NS	0.046
20:0	Arachidic	NS	NS	0.018	NS
16:1n-7	Palmitoleic	NS	0.011	NS	0.019
17:1	Heptadecenoic	NS	NS	NS	NS
18:1n-9	Oleic	NS	NS	NS	NS
18:1n-11	Vaccenic	NS	NS	NS	NS
20:1n-7	Paullinic	NS	NS	NS	0.027
20:1n-9	Gondoic	NS	NS	NS	0.045
20:1n-11	Cetoleic	NS	NS	NS	NS
18:2n-6	Linoleic	0.039	0.009	NS	0.019
20:2n-6	Eicosadienoic	NS	NS	NS	NS
18:3n-6	Linolenic	NS	0.043	0.009	0.011
20:3n-3	Eicosatrienoic	NS	NS	NS	NS
20:4n-6	Arachidonic	NS	NS	NS	NS
ΣSFA		NS	NS	NS	NS
ΣMUFA		NS	0.047	NS	NS
ΣPUFA		NS	NS	NS	NS
ΣUFA		0.029	0.039	NS	NS
ΣPUFAn-3		NS	0.038	0.030	0.008
ΣPUFAn-6		NS	NS	NS	NS
ΣPUFA/SFA		0.037	0.029	NS	NS
ΣSFA/UFA		NS	NS	NS	NS
PUFAn-6/PUFAn-3		NS	0.035	0.041	0.007

NS—*p* > 0.05.

**Table 6 animals-14-01156-t006:** The Pearson correlation coefficient (r) between the level of Zn and the content of fatty acids and cholesterol in the thigh meat.

	ΣSFA	ΣMUFA	ΣPUFA	ΣUFA	ΣPUFAn-3	ΣPUFAn-6	Cholesterol, mg
Zn content, mg	−0.699	−0.560	−0.897	−0.669	−0.959	−0.936	−0.248
*p*-value	0.300	0.440	0.103	0.331	0.041	0.064	0.752

**Table 7 animals-14-01156-t007:** Dietetic values of thigh meat.

	100-Zn-Sulphate *n* = 10	50-Zn-Sulphate *n* = 10	100-Zn-Gly *n* = 10	50-Zn-Gly *n* = 10	SEM	*p*-Value
AIs	0.353 ^a^ ± 0.04	0.364 ^b^ ± 0.03	0.373 ^c^ ± 0.04	0.369 ^b^ ± 0.05	0.02	0.042
TIs	0.720 ^a^ ± 0.08	0.742 ^c^ ± 0.05	0.738 ^c^ ± 0.61	0.730 ^b^ ± 0.01	0.24	0.023
H/H	2.793 ^d^ ± 0.35	2.674 ^b^ ± 0.21	2.652 ^a^ ± 0.30	2.703 ^c^ ± 0.37	0.19	0.021
HPI	2.872 ^d^ ± 0.34	2.760 ^c^ ± 0.20	2.712 ^a^ ± 0.29	2.750 ^b^ ± 0.04	0.16	0.008

Results are presented as the average of three repetitions; ^a–d^—means with different superscripts in lines differ at *p* < 0.05; SEM—standard error of mean; AIs—Atherogenic Indices; TIs—Thrombogenic Indices; H/H—Hypocholesterolemic/Hypercholesterolemic fatty acid ratio; HPI—Health-Promoting Index.

**Table 8 animals-14-01156-t008:** Effect of experimental factor on dietetic value of thigh meat.

	100-Zn-Sulphate vs. 50-Zn-Sulphate	100-Zn-Sulphate vs. 100-Zn-Gly	100-Zn-Gly vs. 50-Zn-Gly	50-Zn-Sulphate vs. 50-Zn-Gly
AIs	0.025	0.019	0.044	NS
TIs	0.039	0.034	0.048	0.043
H/H	0.016	0.009	0.008	0.045
HPI	0.033	0.019	0.032	0.043

NS–*p* > 0.05; AIs—Atherogenic Indices; TIs—Thrombogenic Indices; H/H—Hypocholesterolemic/Hypercholesterolemic fatty acid ratio; HPI—Health-Promoting Index.

## Data Availability

The data presented in this study are available upon request from the corresponding author.

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
