# Peer review of "Fatty Acid Profile and Dietary Value of Thigh Meat of Broiler Chickens Receiving Mineral or Organic Forms of Zn"

_animals, 2024, doi:10.3390/ani14081156_

Round 1
Reviewer 1 Report
Comments and Suggestions for Authors
Comments to the Authors
Title: Fatty acid profile and dietary value of thigh meat of broiler chickens receiving mineral or organic forms of Zn
LN=Line Number
Summary: This manuscript explore the provisional effect of mineral or organic form of Zn on fatty acid profile and the dietary value of thigh meat. This sort of novel studies should always be encouraged keeping the objectives as the findings may be usefully applied (i) in profit maximization through cut down the safety margins of high cost mineral supplements and (ii) through lowering the impact of excretion of excess mineral on environmental pollution which may occur as a result of overestimated animal nutritional requirements. The approach used by the authors and the argument built to meet the stated objectives of the study is well explained and acceptable and can be applied in commercial broiler production.
General comments: Title reflects the study carried out. Summary and abstract adequately describe the study conducted. The manuscript is well written but the authors need to be focused on accurate use of some of the symbols. Some basic information are missed in certain Tables so need additions. The references used in the manuscript are adequate and the most are recent.
Specific comments:
LN100: Please include the details of sex, BW±SD in parenthesis.
LN113: Please mention the nutritional standard you followed as a reference.
LN113: Please provide the details of cage dimensions (L x W x H)
LN115-116: Please use the correct symbol: ℃
LN116: Please provide the details on lighting schedule you have followed and the form (Pellets/mash/crumbles) of the diets.
LN119: Please provide details on slaughtering method used.
LN115-120: Please use the correct symbol: ℃
LN122: Please correct as: -20℃
LN128: Please mention whether in duplicates or triplicates.
LN144: Figure 1: Please provide manufacturer details.
LN144: Figure 1: Nutrient composition: Calculated or analyzed? Please mention.
LN194: Table 1: Please mention whether these values on as fresh or DM basis.
LN195: Table 1: Please state the n here. n=2, n=3 (Sample numbers)
LN219: SEM: should be pooled standard error of mean. Please correct.
LN220: Table 3: Please mention the number of samples you analyzed. n=2 or n=3.
LN277: SEM: should be pooled standard error of mean. Please correct.

Author Response
Reviewer 1
AU: Thank you for the positive feedback. We have marked all the amendments in yellow.
Summary: This manuscript explore the provisional effect of mineral or organic form of Zn on fatty acid profile and the dietary value of thigh meat. This sort of novel studies should always be encouraged keeping the objectives as the findings may be usefully applied (i) in profit maximization through cut down the safety margins of high cost mineral supplements and (ii) through lowering the impact of excretion of excess mineral on environmental pollution which may occur as a result of overestimated animal nutritional requirements. The approach used by the authors and the argument built to meet the stated objectives of the study is well explained and acceptable and can be applied in commercial broiler production.
General comments: Title reflects the study carried out. Summary and abstract adequately describe the study conducted. The manuscript is well written but the authors need to be focused on accurate use of some of the symbols. Some basic information are missed in certain Tables so need additions. The references used in the manuscript are adequate and the most are recent.
AU: Thank you very much for your positive comment.
Specific comments:
LN100: Please include the details of sex, BW±SD in parenthesis.
AU: Thank you for your comment. We wrote “On the first day, two hundred one-day-old Ross 308 cocks were assigned to four ex-perimental groups” (line 99-100) and “Each group consisted of 50 male chickens, placed in 5 cages with 10 chicks in each.” (line 109)
LN113: Please mention the nutritional standard you followed as a reference.
AU: Thank you for your comment. We have corrected the text in accordance with the Reviewer's comment. We wrote “The composition and nutritional value of experimental mixtures (Agropol, Motycz, Poland) are shown in Table 1 and Table 2.” (line 113-114)
LN113: Please provide the details of cage dimensions (L x W x H)
AU: We have corrected in accordance with the Reviewer's comment. We wrote “The broiler chickens were raised in 1-m2 cages until day 42 with unlimited access to feed and water. The cages were 100 cm wide, 100 cm deep, and 75 cm high. They had replaceable grates adapted to the age of the birds and were equipped with 2 nipple drinkers and a feeder (70 cm long, 8.5 cm wide, and 9 cm deep).” (line 115-118)
LN115-116: Please use the correct symbol: ℃
AU: We have corrected in accordance with the Reviewer's comment.
LN116: Please provide the details on lighting schedule you have followed and the form (Pellets/mash/crumbles) of the diets.
AU: We have corrected in accordance with the Reviewer's comment. We wrote “The chickens were provided feed (in pellet form) and water ad libitum.” (line 125-126). We also wrote "The lighting, humidity, and temperature were consistent with the requirements established for Ross 308 chicken rearing. The chickens were kept at a 23/1 light/dark photoperiod from day 0 to day 7; from day 8 onwards, the darkness period was on average approximately 5 hours. The light intensity was 35 lux until rearing day 7 and was reduced to 5-10 lux on day 8. During the dark period, the light intensity was below 0.4 lux.” (line 120-125)
LN119: Please provide details on slaughtering method used.
AU: We have corrected in accordance with the Reviewer's comment. We wrote “ At the end of the experiment, before slaughter, all the chickens were individually weighed after 10-h fasting, and 10 representative chickens with a body weight close to the average value in the group were decapitated after electrical stunning (150 mA, frequency of 200 Hz for 4 s). The chickens were plucked, their viscera were removed, the carcasses were stored overnight at 4 â—¦C, and thigh muscles were collected. After removal of the skin, dissected thigh muscles were placed individually in plastic bags and frozen at -20 °C until chemical analyses were carried out (max. 4 weeks).” (line 127-133)
LN115-120: Please use the correct symbol: ℃
AU: We have corrected in accordance with the Reviewer's comment.
LN122: Please correct as: -20℃
AU: We have corrected in accordance with the Reviewer's comment.
LN128: Please mention whether in duplicates or triplicates.
AU: We have corrected in accordance with the Reviewer's comment. We wrote “All chemical analyses were performed in three replications.” (line…………….)
LN144: Figure 1: Please provide manufacturer details. LN144: Figure 1: Nutrient composition: Calculated or analyzed? Please mention.
AU: Figure 1 has been modified, information about the feed manufacturer and the nutrient composition of mixtures has been included in the text
LN194: Table 1: Please mention whether these values on as fresh or DM basis.
AU: We have corrected in accordance with the Reviewer's comment.
LN195: Table 1: Please state the n here. n=2, n=3 (Sample numbers)
AU: We have corrected in accordance with the Reviewer's comment. We wrote under the table “Results are presented as the average of three repetitions”, and number of samples in the header of all tables
LN219: SEM: should be pooled standard error of mean. Please correct.
AU: We have corrected in accordance with the Reviewer's comment.
LN220: Table 3: Please mention the number of samples you analyzed. n=2 or n=3.
AU: We have corrected in accordance with the Reviewer's comment.
LN277: SEM: should be pooled standard error of mean. Please correct.
AU: We have corrected in accordance with the Reviewer's comment.
AU: Thank you for the positive feedback. Thank you very much for such a positive opinion which is very important to us.
Anna Winiarska-Mieczan
Reviewer 2 Report
Comments and Suggestions for Authors
Journal: Animals
Review comments for Article entitled:
Fatty acid profile and dietary value of thigh meat of broiler chickens receiving mineral or organic forms of Zn
This is an interesting and useful experimental study aimed to investigate two research objectives: (1) whether the specific form of used zinc (Zn) (mineral - sulphate or organic - glycine chelate) added in the diet of 200 Ross 308 broiler chickens affects the chemical composition and dietary qualities of thigh meat, and (2) to determine whether a reduction in Zn supply to 50% of the requirement will affect the chemical composition and dietary qualities of thigh meat (pH, total fat and cholesterol content, and the fatty acid profile, including its indicators: the Atherogenic Index (AI), Thrombogenic Index (TI), Hypocholesterolemic / Hypercholesterolemic (H/H) fatty acid ratio, and the Health-Promoting Index (HPI).
General comments:
The Introduction is well written, in a clear, precise, and systematic manner. The gap of knowledge, as well as the aim of the study are clearly presented and explained.
The experimental designs, sample size, as well as analytical and statistical analyses are appropriate, and the data collection, procession, and interpretation are reproducible based on the details given in five subsections of the Materials and Methods section.
The obtained results are also clearly presented, discussed in systematic manner, presented in seven tables and compared with previously published results.
The Conclusions are well written, and represent the core of your research and the obtained results.
Specific comments:
In order to improve the quality of this manuscript I would suggest the following:
- Lines 45-47: Rephrase this part of the sentence in order to avoid excessive use of term “which”.
- Line 50: How are the unused minerals released and where?
- Line 51: “Our study” change to “Our studies”, as you mentioned two of them.
- Line 67: Introduce the meaning of these abbreviations. I suggest the following: “…but showed no effect of Zn deficiency on saturated fatty acids (SFA) and monounsaturated fatty acids (MUFA) [13].”
- Line 112: “proposed by the [18]” change to “proposed by the National Research Council [18]”.
- Lines 115, 116 and 120: please check the writing of the symbol “°C”.
- Line 170: add software specification, for example: “…using Statistica 13.1 software (TIBCO Software Inc., Palo Alto, CA, USA).
- Lines 182-184: Delete this sentence.
- Tables 2, 4 and 7: Delete the last “vs” in the first row of the last column.
- Line 425: the name of the journal should be in italic and bold the year of the publishing.
Comments on the Quality of English LanguageMinor editing of English language required.
Author Response
Reviewer 2
AU: Thank you for the positive feedback. We have marked all the amendments in green.
This is an interesting and useful experimental study aimed to investigate two research objectives: (1) whether the specific form of used zinc (Zn) (mineral - sulphate or organic - glycine chelate) added in the diet of 200 Ross 308 broiler chickens affects the chemical composition and dietary qualities of thigh meat, and (2) to determine whether a reduction in Zn supply to 50% of the requirement will affect the chemical composition and dietary qualities of thigh meat (pH, total fat and cholesterol content, and the fatty acid profile, including its indicators: the Atherogenic Index (AI), Thrombogenic Index (TI), Hypocholesterolemic / Hypercholesterolemic (H/H) fatty acid ratio, and the Health-Promoting Index (HPI).
General comments:
The Introduction is well written, in a clear, precise, and systematic manner. The gap of knowledge, as well as the aim of the study are clearly presented and explained.
The experimental designs, sample size, as well as analytical and statistical analyses are appropriate, and the data collection, procession, and interpretation are reproducible based on the details given in five subsections of the Materials and Methods section.
The obtained results are also clearly presented, discussed in systematic manner, presented in seven tables and compared with previously published results.
The Conclusions are well written, and represent the core of your research and the obtained results.
AU: Thank you for the positive feedback.
Specific comments:
In order to improve the quality of this manuscript I would suggest the following:
- Lines 45-47: Rephrase this part of the sentence in order to avoid excessive use of term “which”.
AU: Thank you for your comment. We wrote “It is known that the way poultry is fed affects the fatty acid profile of meat [3], and it is important to avoid modifications that could have an adverse effect, particularly for thigh meat which is higher in fat than breast meat, considered to be low-fat [4].” (line 44-47)
- Line 50: How are the unused minerals released and where?
AU: Thank you for your comment. We wrote “Therefore, firstly, it increases the animal breeding cost, and secondly, unused minerals are excreted by animals and hence contribute to environmental pollution.” (line 49-51)
- Line 51: “Our study” change to “Our studies”, as you mentioned two of them.
AU: We have corrected in accordance with the Reviewer's comment.
- Line 67: Introduce the meaning of these abbreviations. I suggest the following: “…but showed no effect of Zn deficiency on saturated fatty acids (SFA) and monounsaturated fatty acids (MUFA) [13].”
AU: We have corrected in accordance with the reviewer's comment.
- Line 112: “proposed by the [18]” change to “proposed by the National Research Council [18]”.
AU: We have corrected in accordance with the Reviewer's comment.
- Lines 115, 116 and 120: please check the writing of the symbol “°C”.
AU: We have corrected in accordance with the Reviewer's comment.
- Line 170: add software specification, for example: “…using Statistica 13.1 software (TIBCO Software Inc., Palo Alto, CA, USA).
AU: We have corrected in accordance with the Reviewer's comment. We wrote “We conducted a statistical analysis of the results using Statistica 13.1 software for Windows (TIBCO Software Inc., Palo Alto, CA, USA, https://www.statsoft.pl/statistica_13/).” (line 192-193)
- Lines 182-184: Delete this sentence.
AU: We have corrected in accordance with the Reviewer's comment.
- Tables 2, 4 and 7: Delete the last “vs” in the first row of the last column.
AU: Thank you for your comment. We have corrected in accordance with the Reviewer's comment.
- Line 425: the name of the journal should be in italic and bold the year of the publishing.
AU: Thank you for your comment. We have corrected in accordance with the Reviewer's comment.
Minor editing of English language required.
AU: Thank you for your comment. The text has been proofread by a native speaker.
AU: Thank you for the positive feedback. Thank you very much for such a positive opinion which is very important to us.
Anna Winiarska-Mieczan
Reviewer 3 Report
Comments and Suggestions for Authors
The study aimed to explore the effects of various levels and forms of Zn supplementation on the fatty acid profiles and dietary value of poultry meat. This investigation sheds light on a relatively understudied area, providing valuable insights into how meat quality may be influenced by minerals like Zn.
General comments:
1. The current study was obviously follows a 2x2 factorial design with Zn levels and forms as two factors. However, the authors used one-way ANOVA and Student–Newman–Keuls test to analyze the data. The Newman–Keuls test is well known to cause Type I error.
The correct way is to analyze the data by two-way ANOVA to so that the authors can analyze the main effect (level and form) and the interaction effect. And if significant difference was detected, Tukey post hoc test should be performed to separate the means instead of the Newman–Keuls test.
The current statistical analysis is not appropriate and authors should redo the analysis.
Specific comments:
1. L109-116, this current description of the treatment allocation information is all over the place and hard to capture. Were there 5 replicates per treatment group with 10 birds per replicate? Please write it in a more clear manner.
2. L110, for reference 7, why did the authors used the Nutrient Specification for Ross 308 Parent stock? The parent stock is completely different from the actual production used broilers in nutrient requirement and management. The authors stated that they used Ross 308 chicks in L99, please check this and refer to the correct guide.
3. Figure 1, Why display the diet formulation in a Figure, it would be way more easier to read in a table. The review suggest the authors put these information in a table instead of in a figure. Also the detailed diet formulation need to be provided.
4. Figure 1, Were these nutrient composition shown in the figure calculated or actual analyzed values? Both values need to be displayed.
5. Figure 1, The crude protein level of the diets were 20.2 for starter; 18.2 for grower, and 18.1 for finisher. However, the breeder recommended levels were 23.0 for starter, 21.5 for grower, and 19.5 for finisher. Was there a deliberate choice by the authors to formulate the diets with lower protein content, and if so, what is the rationale for this reduction in crude protein in the diet?
6. L285-312, the authors spent a large section of the discussion introducing how Zn is related to fatty acid metabolism. As the discussion section is supposed to discuss the results rather than providing background information that should be done in the introduction. The reviewer suggested that the authors reduce the length of these information. If the authors would like to keep these information, they should consider how to integrate them with the observed results to better interpret the observations.
7. L315-321, The reviewer strongly doubts that the authors can infer the sufficiency of zinc solely based on the lack of significant difference in n-6 PUFA.
Author Response
Reviewer 3
AU: Thank you for the positive feedback. We have marked all the amendments in blue.
The study aimed to explore the effects of various levels and forms of Zn supplementation on the fatty acid profiles and dietary value of poultry meat. This investigation sheds light on a relatively understudied area, providing valuable insights into how meat quality may be influenced by minerals like Zn.
General comments:
- The current study was obviously follows a 2x2 factorial design with Zn levels and forms as two factors. However, the authors used one-way ANOVA and Student–Newman–Keuls test to analyze the data. The Newman–Keuls test is well known to cause Type I error. The correct way is to analyze the data by two-way ANOVA to so that the authors can analyze the main effect (level and form) and the interaction effect. And if significant difference was detected, Tukey post hoc test should be performed to separate the means instead of the Newman–Keuls test. The current statistical analysis is not appropriate and authors should redo the analysis.
AU: We would like to express our sincere appreciation to the Reviewer for their highly valuable comment. In each of our conducted experiments, we carefully select the statistical model that best describes our data. However, we concur with the Reviewer that in this particular case, we may not have chosen the optimal statistical model for analyzing our data. We have now included the new results and provided a comprehensive discussion on these findings. We hope that the implementation of our new statistical model will meet the approval of the Reviewer.
As recommended, we have now employed ANOVA to obtain more reliable results. To ensure appropriate analysis, we performed ANOVA with planned comparisons using appropriate contrast coefficients. The normal distribution of variables was assessed using the Shapiro-Wilk test; for data with a normal distribution, one-way analysis of variance (ANOVA) with planned con-trasts (contrast analysis) was applied. The groups were compared based on the form (mineral vs. organic) and content of Zn (100% or 50% of the requirement). Therefore, we performed comparisons between the 100-Zn-sulphate and 100-Zn-Gly groups and be-tween the 50-Zn-sulphate and 50-Zn-Gly groups (estimated effect of the form of Zn), as well as between the 100-Zn-sulphate and 50-Zn-sulphate groups, and between the 100-Zn-Gly and 50-Zn-Gly groups (estimating the effect of the amount of Zn used). This statistical approach allows for a reasonably reliable estimation of the significance of differences between selected groups while minimizing the increase in the α level. If the data did not follow a normal distribution, Kruskal-Wallis tests were conducted.
Specific comments:
- L109-116, this current description of the treatment allocation information is all over the place and hard to capture. Were there 5 replicates per treatment group with 10 birds per replicate? Please write it in a more clear manner.
AU: Thank you for your comment. We wrote: „Each group consisted of 50 male chickens, placed in 5 cages with 10 chicks in each.” (line 109). We hope the text is clearer now.
- L110, for reference 7, why did the authors used the Nutrient Specification for Ross 308 Parent stock? The parent stock is completely different from the actual production used broilers in nutrient requirement and management. The authors stated that they used Ross 308 chicks in L99, please check this and refer to the correct guide.
AU: Thank you for your comment. We verified the references, we made a mistake, it should be Aviagen 2014 - the error was corrected
- Figure 1, Why display the diet formulation in a Figure, it would be way more easier to read in a table. The review suggest the authors put these information in a table instead of in a figure. Also the detailed diet formulation need to be provided.
- Figure 1, Were these nutrient composition shown in the figure calculated or actual analyzed values? Both values need to be displayed.
AU: Figure 1 has been modified. Information on the composition and nutritional value of the experimental mixtures is provided in Tables 1 and 2.
- Figure 1, The crude protein level of the diets were 20.2 for starter; 18.2 for grower, and 18.1 for finisher. However, the breeder recommended levels were 23.0 for starter, 21.5 for grower, and 19.5 for finisher. Was there a deliberate choice by the authors to formulate the diets with lower protein content, and if so, what is the rationale for this reduction in crude protein in the diet?
AU: Thank you for your comment. The mixtures for broiler chickens were balanced using the WinPasze Pro computer program. The total protein content calculated for mixtures according to the NRC standards (1994) was 23.2% in the starter mixture, 20.4% in the grower mixture, and 18.3% in the finisher mixture, while the chemical analysis of ready-made mixtures showed that the protein content in the diets was 20%.2%, 18.2%, and 18.1%, respectively. The level of total protein in seeds and grains depends on a number of factors, mainly the cultivar, weather conditions prevailing during the growing season, and agrotechnological methods used for cultivation. We hope that our explanations will satisfy the Reviewer and clarify the situation.
- L285-312, the authors spent a large section of the discussion introducing how Zn is related to fatty acid metabolism. As the discussion section is supposed to discuss the results rather than providing background information that should be done in the introduction. The reviewer suggested that the authors reduce the length of these information. If the authors would like to keep these information, they should consider how to integrate them with the observed results to better interpret the observations.
AU: Many thanks to the Reviewer for the right comment. We have removed general information on desaturation and fatty acid metabolism from the discussion, as it was not relevant to the scope of the study.
- L315-321, The reviewer strongly doubts that the authors can infer the sufficiency of zinc solely based on the lack of significant difference in n-6 PUFA.
AU: Many thanks to the Reviewer for the right comment. We wrote „In this study, we noted a statistically significant effect of the level of Zn in meat on the content of PUFAn-3, with no significant effect on PUFAn-6, which is one of the factors suggesting that the level of Zn supplied to chickens with feed was not insufficient even when Zn sulphate covered 50% of the standard requirement.” (line 319-322)
AU: Thank you for the positive feedback. Thank you very much for such a positive opinion which is very important to us.
Anna Winiarska-Mieczan
Round 2
Reviewer 3 Report
Comments and Suggestions for Authors
1. L111 and L112, it is very confusing to the reviewer that seemed like the authors used two nutrient requirements one if the Ross 308 specification and the other is NRC recommendation? Why not just use the more current Ross 308 requirements to formulate the diet? Furthermore, the NRC version was published in 1994 which was very outdated.
2. The authors reconducted the statistical analysis although not as the reviewer recommended, the contrast method was indeed another way to effectively reduce the potential error in statistical analysis. But how was the the comparison based on the form (mineral vs. organic) and content of Zn (100% or 50% of the requirement) analyzed, was it also by contrast analysis?
The way the authors presented the results was still a bit confusing and need to be improved:
L211, need to present the p-values here for total content of ash and Zn
L207-214, there is no description on the results of crude ash although significant difference were observed in all four comparisons
L223-241, the reviewer noticed that the authors presented all the statistical results from Table 5 as (p < 0.05) where those in Table 6 in specific numbers? What is the rationale for doing so? The way to present p-values need to be consistent.
L223-241, the results were presented in a very random manner with C17:0, C18:0 described before C10:0 and C12:0. The authors should follow a clear pattern in describing the results such as from short-chain to long chain fatty acid.
L228, the p-value for C8:0 was described in the text but how about those for the C14:0 and C15:0?
L231-232, provide the p-value
L206, throughout the results session, the authors used the phrase such as "statistically confirmed effects of the form or amount of Zn"
What does the author mean by "statistically confirmed"? The results need to be described more specifically such as which form was higher or which concentration of Zn showed higher FA values, the p-values need to be also showed in the text.
P-values are also missing in multiple statements where the authors made to comparison such.
Such issues need to be adjusted throughout the result session.
Author Response
Reviewer 3
AU: Thank you for the positive feedback. We have marked all the amendments in yellow.
- L111 and L112, it is very confusing to the reviewer that seemed like the authors used two nutrient requirements one if the Ross 308 specification and the other is NRC recommendation? Why not just use the more current Ross 308 requirements to formulate the diet? Furthermore, the NRC version was published in 1994 which was very outdated.
AU: We agree with the reviewer's suggestion that providing 2 sources regarding nutrient requirements has caused some confusion. The nutritional content according to the Ross 308 specifications and NRC recommendations does not differ significantly. However, the amount of Zn in the diet was based on the nutritional recommendations for Ross 308 broilers (Aviagen, 2014), i.e., 100 mg/kg of Zn, regardless of its content in the basic diet ingredients. According to these recommendations, the Zn content should be the same in all rearing periods, which was taken into account in our study. Previously recommended Zn amounts according to NRC (1994) for various market broilers, except for growing male Ross 308 chickens, ranged from 14-57 mg/kg. Our results stem from a series of experiments conducted since 2012, when the NRC 1994 norms were widely used in Poland. This experimental setup was approved by the Ethics Committee. The data presented in this manuscript are from an experiment conducted in 2023. In subsequent experiments, we will adjust the norm accordingly. We sincerely thank the reviewer for this invaluable comment.
- The authors reconducted the statistical analysis although not as the reviewer recommended, the contrast method was indeed another way to effectively reduce the potential error in statistical analysis. But how was the the comparison based on the form (mineral vs. organic) and content of Zn (100% or 50% of the requirement) analyzed, was it also by contrast analysis?
AU: All statistical analyses were performed using the contrast method
The way the authors presented the results was still a bit confusing and need to be improved:
L211, need to present the p-values here for total content of ash and Zn
AU: We have corrected in accordance with the Reviewer's comment
L207-214, there is no description on the results of crude ash although significant difference were observed in all four comparisons
AU: We have corrected in accordance with the Reviewer's comment. We wrote „The content of crude ash in meat can be presented as follows: 100-Zn-Gly > 50-Zn-Gly > 100-Zn-sulfate > 50-Zn-sulfate.” (line 212-213)
L223-241, the reviewer noticed that the authors presented all the statistical results from Table 5 as (p < 0.05) where those in Table 6 in specific numbers? What is the rationale for doing so? The way to present p-values need to be consistent.
AU: In Table 6, we presented only the results of the statistical analysis, aiming to illustrate dependencies and trends. Therefore, we deemed that specific values for statistically insignificant results would not be necessary, as they could only complicate the interpretation of the data. In Table 5, on the other hand, specific values regarding the examined meat are provided, hence all values are included. We hope that such an explanation will satisfy the reviewer.
L223-241, the results were presented in a very random manner with C17:0, C18:0 described before C10:0 and C12:0. The authors should follow a clear pattern in describing the results such as from short-chain to long chain fatty acid.
AU: Thank you very much for your comment. We moved the sentences about C18:0 acids to the end of the paragraph. (line 241-243)
L228, the p-value for C8:0 was described in the text but how about those for the C14:0 and C15:0?
AU: We have corrected in accordance with the Reviewer's comment (line 230-231)
L231-232, provide the p-value
AU: We have corrected in accordance with the Reviewer's comment
L206, throughout the results session, the authors used the phrase such as "statistically confirmed effects of the form or amount of Zn"
What does the author mean by "statistically confirmed"? The results need to be described more specifically such as which form was higher or which concentration of Zn showed higher FA values, the p-values need to be also showed in the text.
P-values are also missing in multiple statements where the authors made to comparison such.
Such issues need to be adjusted throughout the result session.
AU: We have corrected in accordance with the Reviewer's comment
AU: Thank you for the positive feedback. Thank you very much for such a positive opinion which is very important to us.
Anna Winiarska-Mieczan